# Replication: Fairness without demographics through Adversarially Reweighted Learning

## Reproducibility Summary

**Scope of Reproducibility**

We test the claim that Adversarially Reweighted Learning (ARL) improves Rawlsian Max-Min fairness for supervised classification compared to previous methods and simple baselines in the case of missing demographic data.

**Methodology**

We completely re-implemented all models and training routines in PyTorch, using the paper and the published code as a reference. We compared our implementation to the one provided by the authors and then reproduced the hyperparameter search as described in the paper using our implementation. In addition, we applied the method to image data, in order to test how well it generalizes across modalities.

Due to the general simplicity of the used models, it was straightforward to implement the code base from scratch and run the experiments. Parts of the experiments were run on a computing cluster with 12 CPUs and parts on Google Colab (GPU). Overall, it took 4 weeks to produce the results, with four people working on it. A complete grid search took 4 hours and producing the final results with fixed hyperparameters 5 hours, of which a single training run of one model on one dataset took about 2 minutes.

**Results**

We could not replicate the advantage of ARL over the investigated baselines. This seems to be mainly due to a better baseline performance than reported in the paper. Our baseline's performance is on average 2.615 standard deviations higher than the authors'. Our ARL results do not deviate significantly from the papers' result, they are on average 0.841 standard deviations higher.

**What was easy**

ARL itself was very easy to implement. We were also able to run the code provided by the authors quite easily. Running the experiments required little computational resources because of the small datasets and models.

**What was difficult**

Pre-processing the data took time because the notebooks provided by the authors contained some errors that we needed to debug. For the replication of the grid search for hyperparameter optimization of all models on all dataset, we had to limit the training duration to a maximum of 5k steps in order to finish all experiments on time.

**Communication with original authors**

We asked the authors about details regarding their training procedure. The authors provided us with the missing details and adapted their GitHub repository as a response to our communication.

# 1 Introduction

As machine learning (ML) systems are increasingly often applied to high-stakes situations, it is becoming more important to ensure that they do not discriminate against groups of individuals based on features such as race and sex. Lahoti et al. present an approach called *Adversarially Reweighted Learning* (ARL) that aims to improve a model's performance measured according to the Rawlsian Max-Min fairness principle by reformulating the training procedure as an adversarial min-max game [1]. While previous work mainly addresses the case where protected features are part of the dataset, ARL does not require such information. As Lahoti et al. point out, such an approach is important because information about group membership is often sensitive and not available in many real-world scenarios, for example due to privacy concerns.

## 1.1 Replication of Main Experiments

The main empirical claim of Lahoti et al. is that ARL outperforms several other methods across several (tabular) datasets in terms of both performance and fairness. Their most important comparison models are Distributionally robust optimization (DRO) [2], inverse probability weighting (IPW) and a simple empirical risk minimization baseline. The focus of our reproducibility study is therefore whether we can replicate an advantage of ARL over these methods.

We implemented all of the methods ourselves in PyTorch [3]. For ARL, IPW and the baseline, we mostly relied on the description in the paper and referred to the authors' TensorFlow [4] code where necessary. For DRO, we adapted the publicly available implementation to our framework. Our own code is available in the supplementary material.

Following the paper, we performed a grid search over hyperparameters. We report the performance of our own implementation on the same datasets as used by the authors, namely UCI Adult, LSAC and COMPAS. Based on our own results and the process of obtaining these, we evaluate the replicability of the paper. We pay special attention to the following criteria:

- Are our own results in line with the results reported by the authors?
- What challenges did we meet when replicating the authors' method?
- What differences exist between the authors' paper and their code?
- Can we support the explicit and implicit assumptions made by the authors to substantiate their method?

## 1.2 Additional Contributions

Given the tabular nature of the UCI Adult, LSAC and COMPAS datasets, the authors apply their method to fully-connected networks. Adversarial reweighting is, however, not limited to fully-connected networks, but serves as a general framework to modify optimization such that worst-case performance over unobserved protected groups is improved. We therefore applied the method to image data, in order to test how well it generalizes across modalities. For this, we trained a convolutional ARL on a custom image dataset based on EMNIST.

# 2 Methodology

## 2.1 Model Descriptions

ARL, DRO and IPW have in common that they modify the optimization objective in order to improve fairness. Even though the ARL introduces an additional adversary network to dynamically learn the exact form of this modification, the network architecture of the inference model is identical for all different methods. (Note that this does not transfer to the adversary network of ARL, which is not used for inference.)

For training on the UCI Adult, LSAC and COMPAS datasets, we used a simple fully-connected feed forward neural network. As in the paper, the network consists of two hidden layers with 64 and 32 units respectively, and ReLU activation functions applied after each hidden layer. For the given binary classification tasks, the output layer consists of one output unit.

**Adversarially Reweighted Learning (ARL)**

The key component to ARL is its adversarial setting. Concretely, a so-called *learner* is trained to minimize a task-specific loss function. Alternatingly, a so-called *adversary* is trained to reweight the loss terms that are produced by the

learner for each individual training sample in order to maximize the overall loss. Using $\theta$ and $\phi$ to respectively denote the parameters of the learner and the adversary, the overall training objective for $n$ training samples is formulated as:

$$J(\theta, \lambda) := \min_\theta \max_\phi \sum_{i=1}^n \lambda_\phi(x_i, y_i)\ell(h_\theta(x_i), y_i),$$

where $\ell : \hat{Y} \times Y \to \mathbb{R}$ denotes the loss function of the underlying task and $h_\theta : X \to \hat{Y}$ denotes the function that is implemented by the learner. The weights $\lambda_\phi(X, Y)$ are produced by the adversary as follows:

$$\lambda_\phi(x_i, y_i) = 1 + n \cdot \frac{f_\phi(x_i, y_i)}{\sum_{j=1}^n f_\phi(x_j, y_j)},$$

where $f_\phi(x_i, y_i)$ is the adversary's output. During training, the adversary implicitly learns to identify areas in the input space that lead to high losses $\ell$. By assigning higher weights to samples from these areas, it drives the learner towards improving in the identified subspaces. The authors claim that this ultimately leads to improved performance for protected groups whose sensible characteristic features are not included in the input data $X$, i.e. the groups that are identified by the adversary correspond to a certain degree to the groups that are to be protected.

We implemented the adversary with one linear layer, followed by a sigmoid function and the normalization, scaling and shifting as stated above. Instead of using the sum of single reweighted losses, we used the mean, following the authors' code. In their code they used two consecutive linear layers, which is not perfectly in line with the paper. Their code also reveals that the learner network was pretrained, which they do not mention in their paper.

**Empiricial Risk Minimization (Baseline)**

This model is identical to the learner in ARL. The authors note that they increased the number of hidden units in this model to compensate for the added capacity in form of the adversary in ARL. However, it was neither clear from the paper how they implemented this, nor evident from their code that they actually did. Going by parameter count, a single additional unit would already result in more parameters than for ARL. Since the simple fully-connected model did already match the performance of ARL (see Table 1), we did not investigate the effect of adding more units. The fully-connected model will also be referred to as 'baseline' in the rest of the text.

**Inverse Probability Weighting (IPW)**

Instead of learning the weights to reweight the loss function dynamically, a simpler approach is to weight each sample with the inverse probability of drawing this sample from the dataset. This gives underrepresented groups in the dataset a larger weight. In the IPW(S) approach, the probabilities are calculated based only on the membership to protected groups (e.g. "sex" or "race"), resulting in weights of the form $1/p(s)$. Additionally to the protected group we can take into account the label, resulting in a model called IPW(S+Y). Here, the probabilities are calculated as the joint probabilities of being in a certain protected group and belonging to a certain label (e.g. "being black" *and* "earning more than fifty thousand dollars a year"). This results in weights of the form $1/p(s, y)$.

IPW can only be used if group probabilities are known. It therefore offers a comparison of the ARL's claim of improving fairness *without* demographics to a method that improves fairness *with* demographics.

**Distributionally Robust Optimization (DRO)**

Similar to ARL, DRO aims to improve Rawlsian Max-Min Fairness for tasks where protected groups are unknown. Hashimoto et al. propose the following optimization objective:

$$\min_{\theta \in \Theta} \mathbb{E}_P \left[\ell(\theta; Z) - \eta\right]_+^2$$

$P$ denotes the data generating distribution, $\ell(\theta; Z)$ the loss of a query $Z$ under the model with parameters $\theta$, and $\eta$ a hyperparameter to threshold the losses. In the publicly available code by Hashimoto et al. that we used in our implementation, a ReLU activation function is applied after subtracting $\eta$ from losses of single data points, ensuring that only positive losses remain. The method focuses on the worst-case group by only considering losses for optimization that exceed the threshold $\eta$, which implicitly decides about the size of the worst-case group. Squaring the thresholded losses upweights the influence of higher losses on the learning objective further.

## 2.2 Datasets

We use the three datasets also used in the paper: UCI Adult [5], COMPAS [6] and LSAC [7], all of which are publicly available. Every dataset describes a binary prediction task. In UCI Adult, the income of a person is to be predicted to be above fifty thousand dollars. COMPAS is concerned with predicting the recidivism of indviduals convicted of crimes. For LSAC, the task is to predict whether a given law student will pass the bar exam.

All datasets exhibit some form of imbalance and include information about sensitive attributes like sex and race which makes them suitable for evaluating model fairness. UCI Adult consists of 48,842 records, of which 67% are male and 85% white (10% black). The dataset is also imbalanced in its target class, as only 24% have an income of more than fifty thousand dollars. LSAC consists of 26,551 samples, of which 56% are male and 83% white (6% black). COMPAS consists of 7,185 records with 81% being male and 37% white (51% black). In all datasets, there exists some correlation between sensitive attributes and target class [8, 7, 9].

**Pipeline**

We split each dataset randomly into 70% training data and 30% test data, except for UCI Adult, where a split is already provided. For our main experiments, the sensitive columns race and sex were removed from the features, such that these demographics were not available to the models. To access sensitive information during evaluation, each data element was categorized as being a member of one of the following groups: "not black and male", "not black and female", "black and male", "black and female". Of these, the group with the fewest members is referred to as minority. This treatment of the subgroups was inferred from the publicly available code. The paper however used the term "white" instead of "not black", which does not accurately reflect the work done in their code. We will adopt their term usage from hereon for better comparability.

We represented categorical features using one-hot encoding and normalized numerical features to have zero mean and unit standard deviation. All features of a datapoint were then appended into one feature vector. A script to download and preprocess the datasets is available as part of our code base.

## 2.3 Hyperparameters

We determined the optimal batch size and learning rate for each method using a grid search and 5-fold cross validation over the same search space as described in the appendix to the paper.

All results were obtained by training from scratch with the obtained optimal hyperparameters using ten different random seeds, then evaluating on the test set and averaging the results. We pretrained the learner for ARL for 250 steps, which is the default in the code provided by the authors. We used AdaGrad as the optimizer, following the paper. Initialization for the weights and the optimizer were chosen to agree with TensorFlow defaults. In the initial experiments, we ran each grid search for 5000 training steps, with early stopping if the overall AUC on the validation set had not improved for 10 epochs.

For the grid search space and the optimal hyperparameters found, see Appendix C. For the hyperparameters obtained by the authors and a comparison to the hyperparameters found in our grid search, see Appendix D.

## 2.4 Computational Requirements

The grid searches took about 3-4h on twelve CPUs for the 5k step runs on tabular data and an additional 10h on a GPU for the image grid searches. The 100k step grid seach took much longer, about 1-2 days. The final results (training each model on ten different random seeds) were obtained from a notebook that ran about 5h on Google Colab with a GPU.

# 3 Results

Table 1 shows the results of comparing ARL to DRO and the baseline model. We observe that all methods obtain very similar results for each dataset and metric. Many of the differences that do exist between methods are not significant, and in any case which methods performs best varies strongly between metrics and datasets.

Table 2 shows the relative deviation of our results in Table 1 to those reported in the paper. Positive numbers mean that our performance was higher, negative that it was lower than that from the paper. All numbers are measured in multiples of the standard error of the difference, calculated as $\sqrt{\sigma_{\text{ours}}^2 + \sigma_{\text{theirs}}^2}$, where $\sigma_{\text{ours}}$ and $\sigma_{\text{theirs}}$ are the standard deviations of our and their results over the 10 runs with different seeds. Because Lahoti et al. do not report standard deviations for macro-avg, minimum and minority AUC, we calculated the errors for those measure using error propagation from the

values they report for all protected subgroups (tables 6 to 8 in their supplementary material). Overall, all three methods tend to have a slightly better performance in our results than in the paper. But for ARL, this effect is smaller than for the baseline and in particular than for DRO. This explains why in our results, ARL doesn't have an advantage like it does in the paper: it doesn't perform any worse, but the comparison methods perform better.

| Dataset | Method | AUC avg | AUC macro-avg | AUC min | AUC minority | Accuracy |
|---|---|---|---|---|---|---|
| Adult | baseline | $0.9093 \pm 0.0016$ | $0.9193 \pm 0.0011$ | $0.8842 \pm 0.0019$ | $\mathbf{0.9440 \pm 0.0040}$ | $0.8498 \pm 0.0070$ |
| Adult | DRO | $0.9102 \pm 0.0007$ | $0.9187 \pm 0.0018$ | $\mathbf{0.8853 \pm 0.0007}$ | $0.9414 \pm 0.0059$ | $\mathbf{0.8574 \pm 0.0015}$ |
| Adult | ARL | $\mathbf{0.9104 \pm 0.0006}$ | $\mathbf{0.9196 \pm 0.0012}$ | $0.8852 \pm 0.0008$ | $0.9433 \pm 0.0042$ | $0.8530 \pm 0.0036$ |
| LSAC | baseline | $0.8309 \pm 0.0061$ | $0.8263 \pm 0.0063$ | $0.8096 \pm 0.0073$ | $0.8371 \pm 0.0087$ | $0.8604 \pm 0.0047$ |
| LSAC | DRO | $0.8242 \pm 0.0051$ | $0.8217 \pm 0.0053$ | $0.8024 \pm 0.0055$ | $0.8361 \pm 0.0088$ | $0.8570 \pm 0.0057$ |
| LSAC | ARL | $\mathbf{0.8333 \pm 0.0046}$ | $\mathbf{0.8272 \pm 0.0041}$ | $\mathbf{0.8115 \pm 0.0057}$ | $\mathbf{0.8375 \pm 0.0042}$ | $\mathbf{0.8644 \pm 0.0032}$ |
| COMPAS | baseline | $0.7357 \pm 0.0025$ | $0.7340 \pm 0.0027$ | $0.6995 \pm 0.0029$ | $0.7475 \pm 0.0024$ | $\mathbf{0.6735 \pm 0.0057}$ |
| COMPAS | DRO | $\mathbf{0.7361 \pm 0.0030}$ | $\mathbf{0.7347 \pm 0.0032}$ | $\mathbf{0.6997 \pm 0.0035}$ | $\mathbf{0.7533 \pm 0.0044}$ | $0.6688 \pm 0.0071$ |
| COMPAS | ARL | $0.7315 \pm 0.0049$ | $0.7290 \pm 0.0059$ | $0.6951 \pm 0.0070$ | $0.7511 \pm 0.0066$ | $0.6498 \pm 0.0380$ |

Table 1: Main results: Baseline vs ARL vs DRO, best results for each dataset are marked bold

| Dataset | Method | AUC avg | AUC macro-avg | AUC min | AUC minority |
|---|---|---|---|---|---|
| Adult | baseline | **6.880** | **8.076** | 0.379 | **2.822** |
| Adult | DRO | **14.134** | **5.102** | **17.038** | 1.408 |
| Adult | ARL | **3.096** | 1.115 | **2.962** | 0.093 |
| LSAC | baseline | **2.702** | **2.113** | **2.456** | 1.287 |
| LSAC | DRO | **22.782** | **21.160** | **21.896** | **12.003** |
| LSAC | ARL | 1.704 | 1.147 | 1.405 | 0.283 |
| COMPAS | baseline | **-2.851** | 1.665 | **8.197** | **-2.352** |
| COMPAS | DRO | **6.439** | **10.727** | **6.219** | **3.881** |
| COMPAS | ARL | **-2.034** | 0.553 | **3.197** | **-3.425** |

Table 2: Relative deviations between our results and those reported in the paper (difference divided by standard deviation across runs, see main text for details). Deviations of more than 2 standard deviations are marked bold. Accuracy was not reported by the authors and is thus missing.

We also ran the code provided by the authors with their optimal hyperparameters (results in Appendix A). There, we get results much closer to those reported in the paper and in particular a (very slight) advantage for ARL. Additionally, we directly compared our Pytorch implementation with the original Tensorflow implementation (Appendix B) and found that our implementation tended to perform somewhat better. We will discuss these various discrepancies further below.

Since some training runs did not end via early stopping but rather due to the maximum number of training steps being reached, we repeated the grid search and evaluation with ten seeds using a maximum number of 100k instead of 5k steps. We did not observe a significant increase in performance for any of the used methods, so we can probably rule out that training for too few training steps distorted our results. The exact results for 100k training steps can be found in Table 11. In the remainder of the main body of this report we use 5k training steps if not stated otherwise.

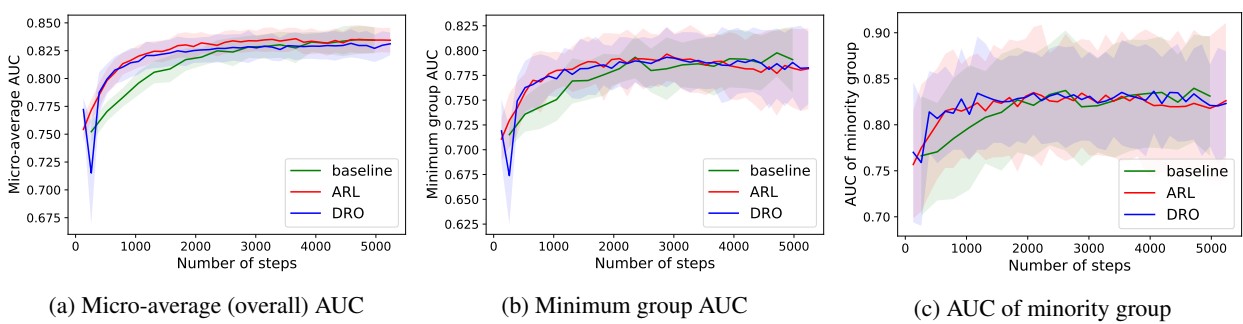

(a) Micro-average (overall) AUC      (b) Minimum group AUC      (c) AUC of minority group

Figure 1: Different AUC values during training of ARL, baseline and DRO on the validation set of LSAC. Average and standard deviation over 10 random seeds.

Figure 1 shows the micro-average AUC, minimum group AUC and the AUC of the minority group of ARL, baseline and DRO on the validation split of the LSAC dataset. As can be seen, ARL shows no consistent advantage in the minimum

group AUC and minority AUC, and a neglectable advantage in the micro-average AUC. Figures 1b and 1c show that, depending on the point in time where the models are saved, there exist some points of comparison where ARL seems to outperform the other models. Given these fluctuations, it is possible that the differences between the authors' and our results can be explained by different points of comparison. We used early stopping and compared the best-performing models, whereas the authors used a fixed number of training steps.

In Table 3, we compare the results between ARL and the naive reweighting approach IPW, either using only protected groups (S) or using protected groups and labels (S+Y) to compute the inverse probability weights. Similarly to the results in Table 1, ARL does not show an advantage over other methods on any of the datasets.

| Dataset | Method | AUC avg | AUC macro-avg | AUC min | AUC minority | Accuracy |
|---|---|---|---|---|---|---|
| Adult | ARL | $0.9104 \pm 0.0006$ | $0.9196 \pm 0.0012$ | $0.8852 \pm 0.0008$ | $0.9433 \pm 0.0042$ | $0.8530 \pm 0.0036$ |
| Adult | IPW(S) | $0.9085 \pm 0.0004$ | $0.9179 \pm 0.0007$ | $0.8826 \pm 0.0005$ | $0.9434 \pm 0.0018$ | $\mathbf{0.8557 \pm 0.0010}$ |
| Adult | IPW(S+Y) | $\mathbf{0.9110 \pm 0.0009}$ | $\mathbf{0.9209 \pm 0.0020}$ | $\mathbf{0.8859 \pm 0.0010}$ | $\mathbf{0.9465 \pm 0.0047}$ | $0.7428 \pm 0.0160$ |
| LSAC | ARL | $0.8333 \pm 0.0046$ | $0.8272 \pm 0.0041$ | $0.8115 \pm 0.0057$ | $\mathbf{0.8375 \pm 0.0042}$ | $\mathbf{0.8644 \pm 0.0032}$ |
| LSAC | IPW(S) | $0.8147 \pm 0.0066$ | $0.8088 \pm 0.0056$ | $0.7933 \pm 0.0089$ | $0.8162 \pm 0.0090$ | $0.8483 \pm 0.0033$ |
| LSAC | IPW(S+Y) | $\mathbf{0.8371 \pm 0.0038}$ | $\mathbf{0.8302 \pm 0.0030}$ | $\mathbf{0.8150 \pm 0.0054}$ | $0.8374 \pm 0.0072$ | $0.8352 \pm 0.0108$ |
| COMPAS | ARL | $0.7315 \pm 0.0049$ | $0.7290 \pm 0.0059$ | $0.6951 \pm 0.0070$ | $0.7511 \pm 0.0066$ | $0.6498 \pm 0.0380$ |
| COMPAS | IPW(S) | $0.7300 \pm 0.0040$ | $0.7294 \pm 0.0058$ | $0.6914 \pm 0.0024$ | $0.7399 \pm 0.0124$ | $0.6661 \pm 0.0159$ |
| COMPAS | IPW(S+Y) | $\mathbf{0.7362 \pm 0.0029}$ | $\mathbf{0.7346 \pm 0.0031}$ | $\mathbf{0.6998 \pm 0.0029}$ | $\mathbf{0.7526 \pm 0.0043}$ | $\mathbf{0.6733 \pm 0.0096}$ |

Table 3: Comparison ARL vs IPW for 5k maximum training steps

## 3.1 Computational Identifiability

To investigate whether a linear adversary is actually capable of identifying protected groups based on the observed features and labels, the authors trained a linear model to predict race or sex based on the features and the labels. They did not provide further details about their training method. Since we only want to test whether the adversary is in principle capable of performing this task and precise accuracies are thus not important, we limit ourselves to a single run.

Training, validation and testing are performed with the optimal hyperparameters found via the grid search in the previous section. The obtained test accuracies for predicting race or sex are shown in Table 4. The results are broadly consistent with the results from the original paper.

|  |  | Adult | LSAC | COMPAS |
|---|---|---|---|---|
| Race | total | 0.907 | 0.943 | 0.618 |
|  | White | 0.997 | 0.988 | 0.603 |
|  | Black | 0.058 | 0.325 | 0.631 |
| Sex | total | 0.842 | 0.585 | 0.802 |
|  | Male | 0.876 | 0.807 | 1.000 |
|  | Female | 0.774 | 0.298 | 0.000 |

Table 4: Accuracies of a linear model trained to predict race or sex.

The per-group accuracies show that in many cases, the linear model achieves its performance by concentrating on the majority class, or even just always predicting the majority class. This raises doubts on whether the adversary is indeed able to adequately identify and upweight minority groups.

## 3.2 Adversary Outputs

To inspect whether the adversary learns meaningful weights, the authors plotted the weights returned by the adversary on the training set of UCI Adult. We evaluate the adversary on the test set of UCI Adult to see whether it has meaningful weights on the test set as well. We perform kernel density estimation with a Gaussian kernel with a bandwidth parameter of 0.3 on the test weights to produce continuous distributions over $\lambda$. Results obtained using scikit-learn [10] are shown in Figure 2.

The computed densities roughly match the authors' results. Thus, we can support their claim that ARL learns to upweight not only misclassified examples, but also underrepresented classes (bottom-right quadrant of Figure 2. Recall that the UCI Adult dataset exhibits a strong class imbalance and as we can see the adversary places larger weights on the less frequent class.

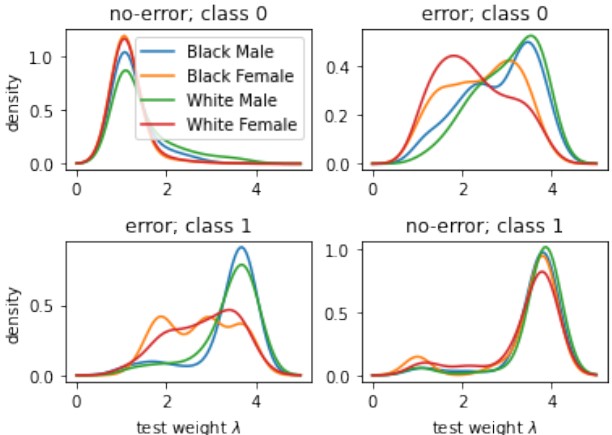

Figure 2: Output of the adversary on the Adult test set.

| Dataset | Method | AUC avg | AUC macro-avg | AUC min | AUC minority | Accuracy |
|---|---|---|---|---|---|---|
| EMNIST_35 | baseline | **0.9507 ± 0.0037** | **0.9436 ± 0.0041** | **0.9256 ± 0.0055** | **0.9256 ± 0.0055** | **0.8778 ± 0.0059** |
| EMNIST_35 | ARL | 0.9370 ± 0.0044 | 0.9281 ± 0.0052 | 0.9051 ± 0.0072 | 0.9051 ± 0.0072 | 0.8616 ± 0.0057 |

Table 5: Baseline vs ARL on image data, best results for each dataset are marked bold. Models were trained for 5k steps or until the early stopping criterion was reached. Reported numbers are averages over 10 seeds.

It is important to point out that ARL apparently *does not* place higher weights on underrepresented demographics (see the orange and red curves in Figure 2). This is in line with the results from Lahoti et al. but does not agree with the original motivation for ARL.

## 4   Extension of the Presented Method

Not just tabular data, but also other modalities are in principle prone to issues regarding fairness. This is why an evaluation of ARL on a different type of data marks a straightforward extension to the analyses presented by Lahoti et al. In this section, we present and evaluate a simple extension of ARL to image data.

### 4.1   Dataset

We resorted to building a custom dataset for the evaluation in order to meet our limited computational resources and maintain the ability to easily tailor the input data to our experiments. The dataset is based on balanced EMNIST [11], a character classification dataset similar to MNIST [12] containing a balanced set of 47 different greyscale characters. We customized the dataset, which we refer to as EMNIST_35, such that the task is a binary prediction with balanced labels, and artificially created one protected minority subgroup. The minority is represented by 35% of the data for both labels. Further details and different data configurations are presented in Appendix G.1.

### 4.2   Model Architecture

Both the learner of ARL and the baseline model contain a single convolutional layer with 64 filters and a $3 \times 3$ kernel with stride 1. Subsequently, a max-pooling layer with kernel size $2 \times 2$ with stride 1 is employed. The adversary uses a similar setup with 32 filters. Both learner and baseline then use two fully connected layers with 64 and 32 neurons with a ReLU activation function in-between. In the adversary, the sample label is appended to the flattened output of the convolutional section and the model output is produced as a linear combination of the resulting vector for each sample. The training procedure and hyperparameters of all image models are further described in Appendix G.3.

### 4.3   Results and Analysis

The model performances regarding the different AUC metrics and the classification accuracy are listed in Table 5. The baseline clearly outperforms ARL on all evaluation criteria. Further experiments showed that the adversary is unable to identify protected groups in the adversarial setting and tends to weight few individual samples with very high weights.

Since the architecture for the learner and the baseline is the same, we further considered the influence of the adversary by investigating if it is capable of identifying the protected groups in a supervised setting. It turns out that the adversary resorts to predicting the majority class in most cases. We ran several experiments with different architectures for the adversary but were not able to produce a performance boost over the reference models similar to those reported by Lahoti et al. on tabular data. Details regarding the results of our experiments are given in Appendix G.

## 5   Discussion

In this report we re-implemented the *Adversarially Reweighted Learning* (ARL) framework by Lahoti et al. in PyTorch. This framework promises a general approach to dealing with group fairness if demographic data are lacking or unknown. However, as shown in Tables 1 and 3 and the analyses that followed, we could not replicate the advantage of ARL over the investigated baselines. This seems to be mainly due to a better baseline performance than in Lahoti et al. As displayed in Table 2, our ARL results are not deviating significantly (mostly $< 3\sigma$) from the papers' result, but our baselines are considerably better than in the original paper.

One possible cause for this is that we used early stopping, whereas the authors of the original paper didn't, as far as their code and paper suggest. This could mean that the number of training steps (which was fixed across methods)

happened to work better for ARL than it did for the other methods. In our setup, the training duration is effectively chosen optimally for each method, which in our view makes for a fairer comparison.

A potential reason why ARL doesn't outperform a simple risk-minimizing baseline is that ARL apparently does not actually upweight minorities, as we showed in Figure 2. As mentioned, this is entirely consistent with the figures in the original paper but does not agree with the motivation for ARL. Perhaps the linear adversary is too weak to be effective – our identifiability results in Table 4 suggest that it mainly learns to exploit the class imbalance. We also point out that the advantage of ARL reported in the original paper is not particularly large, neither in absolute terms nor compared to the errors. This reinforces the idea that the difference between our and the original results is caused by relatively small changes in training procedure, such as the use of early stopping.

Still, ARL is theoretically appealing and our results don't rule out that it might have benefits in other circumstances. Perhaps the datasets chosen by the authors are simply not hard enough to differentiate between the various methods: the baseline model is already quite fair towards subgroups, not leaving much space for ARL to show any differential advantage. Experiments by Zhong suggest that reweighting and oversampling, both methods related to ARL, are not effective at improving fairness in these datasets [13], which further supports our impression of the datasets not being suitable. Our experiments with ARL on image data show that such an extension is not trivial, at the very least. Including an adversary lead to deteriorating performance compared to the baseline which could be explained by real-world images generally containing large amounts of noise. An extensive investigation regarding this and other potential sources of error are left as future work.

On the positive side, we found it relatively easy to run the code provided by the authors and to re-implement their method. There were some issues regarding the dataset preparation and slight differences between the description of the algorithms in the paper and their actual implementation in the code, but overall, none of these posed serious obstacles to replicating the paper.

In conclusion, we did not find an advantage of ARL over baseline models, neither for the datasets used by the authos nor for image data. Given that the results of the original paper favor ARL only very slightly and that the adversary apparently does not weight minorities higher, we do not believe that ARL in its current form can improve fairness on these datasets. However, since the theoretical motivation appears sound, it may be promising to apply ARL to datasets where baselines suffer from more severe fairness issues, or to further investigate whether the capacity of the adversary can be adjusted to improve performance.

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

## A Results using the original setup

We used the code and the optimal hyperparameters provided by the authors to reproduce their experiment. This should yield results very close to those reported in the original paper, though slight deviations are to be expected since the original code doesn't use random seeds. We again averaged our results over 10 runs with different random seeds and calculated the standard deviation over those runs as the error. The error for the authors' results was again calculated based on the values reported in Appendix 8.2 of the original paper. Because the paper doesn't report any errors for the IPW results, we only compare the baseline and ARL results here. Furthermore, we only use the AUC metrics, since the paper doesn't report any accuracies.

Table 6 shows the results we obtained using this setup. In Table 7, we show the relative deviations of these results to those obtained in the paper. Deviations that are significant at the $2\sigma$ level are bolded.

While most of the results are compatible, there are a few with significant differences (sometimes our results are worse and sometimes better than those reported in the paper). We are uncertain what causes these differences – perhaps a slight difference in the data preparation, since the provided notebook didn't run without errors and we had to implement parts of it ourselves.

However, as Table 6 shows, these deviations don't change the overall conclusion: just as in the paper, ARL outperforms the basline for the Adult and LSAC datasets, while tending to perform worse on the COMPAS dataset. In most cases, the differences between baseline and ARL are relatively small, and sometimes not even significant at the $1\sigma$ level.

| Dataset | Method | Micro-avg AUC | Macro-avg AUC | Min AUC | Minority AUC |
|---|---|---|---|---|---|
| Adult | baseline | $0.8956 \pm 0.0007$ | $0.8930 \pm 0.0017$ | $0.8751 \pm 0.0008$ | $0.8883 \pm 0.0058$ |
| Adult | ARL | $\mathbf{0.9041 \pm 0.0008}$ | $\mathbf{0.9152 \pm 0.0018}$ | $\mathbf{0.8781 \pm 0.0010}$ | $\mathbf{0.9440 \pm 0.0073}$ |
| LSAC | baseline | $0.8127 \pm 0.0022$ | $0.8035 \pm 0.0025$ | $0.7891 \pm 0.0033$ | $0.8114 \pm 0.0077$ |
| LSAC | ARL | $\mathbf{0.8204 \pm 0.0027}$ | $\mathbf{0.8174 \pm 0.0056}$ | $\mathbf{0.7990 \pm 0.0048}$ | $\mathbf{0.8224 \pm 0.0145}$ |
| COMPAS | baseline | $\mathbf{0.7322 \pm 0.0022}$ | $\mathbf{0.7312 \pm 0.0028}$ | $0.6943 \pm 0.0038$ | $\mathbf{0.7431 \pm 0.0036}$ |
| COMPAS | ARL | $0.7314 \pm 0.0013$ | $0.7285 \pm 0.0027$ | $\mathbf{0.6947 \pm 0.0048}$ | $0.7370 \pm 0.0076$ |

Table 6: Our results using the original code and hyperparameters, averaged over 10 runs with different seeds. Errors are the standard deviations over those runs.

| Dataset | Method | Micro-avg AUC | Macro-avg AUC | Min AUC | Minority AUC |
|---|---|---|---|---|---|
| Adult | baseline | **-2.698** | 1.370 | 0.004 | 0.538 |
| Adult | ARL | **-2.376** | -0.524 | -1.850 | 0.130 |
| LSAC | baseline | -0.101 | **-2.691** | -0.184 | -1.334 |
| LSAC | ARL | -0.540 | -0.159 | -0.005 | -0.404 |
| COMPAS | baseline | **-3.833** | 0.914 | **5.144** | **-2.676** |
| COMPAS | ARL | **-3.888** | 0.785 | **3.510** | **-4.525** |

Table 7: Relative deviation (in standard deviations) between our reproduced results (see Table 6) and the results reported in the paper. Positive means that ours are higher.

## B Comparison of Implementations

We compared the Tensorflow implementation by the original authors to our Pytorch implementation using exactly the same settings, to check whether they are equivalent. Our results can be found in Table 8. There are some significant differences, even though we took care to use Tensorflow defaults for initialization scheme and optimizer settings. One remaining difference is that the authors implemented the linear layer of the adversary as two consecutive linear layers. This doesn't change the expressiveness but does affect the training dynamics. However, this is only relevant for ARL, and there are significant differences even in the baseline model.

While we were unable to find the cause for these deviations, our implementation tends to perform sightly better in cases where there is any difference at all. So it is unlikely that we made an implementational error that would affect the validity of our results.

| Dataset | Method | AUC avg | AUC minority | Accuracy |
|---|---|---|---|---|
| Adult | baseline | **11.158** | **9.240** | -0.695 |
| Adult | ARL | **11.221** | **7.640** | **2.303** |
| Adult | IPW(S) | **6.796** | **11.547** | -0.424 |
| LSAC | baseline | **6.443** | **6.544** | -0.463 |
| LSAC | ARL | **8.425** | **5.162** | -0.535 |
| LSAC | IPW(S) | **2.752** | **7.051** | -0.635 |
| COMPAS | baseline | 1.891 | **2.124** | 0.440 |
| COMPAS | ARL | **5.106** | 1.691 | 1.171 |
| COMPAS | IPW(S) | **4.376** | 0.429 | 1.414 |

Table 8: Relative deviation between results with our PyTorch implementation and the TensorFlow implementation by the authors. All results are based on runs with 100 steps, a learning rate of 0.1 and a batch size of 128. Positive numbers mean that our implementation achieved higher performance. All numbers are in multiples of the standard error of the difference (calculated as for Table 2, see Section 3 for details). Deviations of more than two standard errors are bold. Since the authors' code does not save macro-avg AUC nor min AUC, these metrics are missing.

## C   Optimal Hyperparameters

We performed grid search to determine optimal hyperparameters. The search space for (batch size, primary learning rate, adversary learning rate, $\eta$) consisted of the following values:

$$[32, 64, 128, 256, 512] \times [0.001, 0.01, 0.1, 1, 2, 5] \times [0.001, 0.01, 0.1, 1, 2, 5] \times [0.5, 0.6, 0.7, 0.8, 0.9, 1],$$

where in the case of methods other than DRO, the $\eta$ parameter was not searched over and in case of methods other than ARL, the adversary learning rate was not searched over. Table 9 shows the optimal hyperparameters for each dataset and model combination, when optimized for overall AUC.

| Dataset | Method | batch size | primary learning rate | adversary learning rate | $\eta$ |
|---|---|---|---|---|---|
| Adult | baseline | 512 | 2 | - | - |
| Adult | DRO | 128 | 1 | - | 0.5 |
| Adult | ARL | 512 | 1 | 0.01 | - |
| Adult | IPW(S) | 128 | 0.01 | - | - |
| Adult | IPW(S+Y) | 256 | 0.1 | - | - |
| LSAC | baseline | 64 | 0.1 | - | - |
| LSAC | DRO | 128 | 1 | - | 0.6 |
| LSAC | ARL | 128 | 0.1 | 0.001 | - |
| LSAC | IPW(S) | 256 | 0.1 | - | - |
| LSAC | IPW(S+Y) | 64 | 0.1 | - | - |
| COMPAS | baseline | 256 | 0.1 | - | - |
| COMPAS | DRO | 256 | 1 | - | 0.6 |
| COMPAS | ARL | 128 | 1 | 0.001 | - |
| COMPAS | IPW(S) | 256 | 0.1 | - | - |
| COMPAS | IPW(S+Y) | 512 | 0.1 | - | - |

Table 9: Hyperparameters with the best overall AUC for each dataset and training method. Models were trained for 5k steps or until the early stopping criterion was reached.

## D   Comparison to authors' hyperparameters

The authors report different optimal hyperparameters than those that we found. We first reran our experiments using their hyperparameters. The results are given in Table 10. They are comparable and if anything slightly worse than those we obtained with our hyperparameters, and ARL does not have any advantage there either.

To understand why the reported hyperparameters differ, we also investigated how important the choice of hyperparameters is after all.

Figure 3 shows a histogram of how often each AUC occured during the grid search. For Adult and COMPAS, we can see that most hyperparameter combinations yield essentially equally good results, all close to the top performance. For

| | | | | | | |
|---|---|---|---|---|---|---|
| Adult | baseline | **0.9105 ± 0.0004** | **0.9190 ± 0.0006** | **0.8855 ± 0.0006** | 0.9409 ± 0.0022 | **0.8582 ± 0.0011** |
| Adult | ARL | 0.9074 ± 0.0008 | 0.9176 ± 0.0009 | 0.8819 ± 0.0012 | **0.9437 ± 0.0021** | 0.8433 ± 0.0030 |
| LSAC | baseline | **0.8328 ± 0.0049** | **0.8276 ± 0.0047** | **0.8114 ± 0.0063** | **0.8365 ± 0.0039** | **0.8616 ± 0.0047** |
| LSAC | ARL | 0.8284 ± 0.0035 | 0.8218 ± 0.0032 | 0.8056 ± 0.0045 | 0.8277 ± 0.0106 | 0.8588 ± 0.0049 |
| COMPAS | baseline | 0.7316 ± 0.0036 | 0.7290 ± 0.0044 | 0.6967 ± 0.0038 | 0.7487 ± 0.0031 | **0.6753 ± 0.0052** |
| COMPAS | ARL | **0.7333 ± 0.0048** | **0.7317 ± 0.0067** | **0.6971 ± 0.0040** | **0.7491 ± 0.0062** | 0.6752 ± 0.0041 |

Table 10: Results with our Pytorch implementation but the optimal hyperparameters reported by the authors. As always, averaged over 10 runs with different seeds, errors are standard deviations.

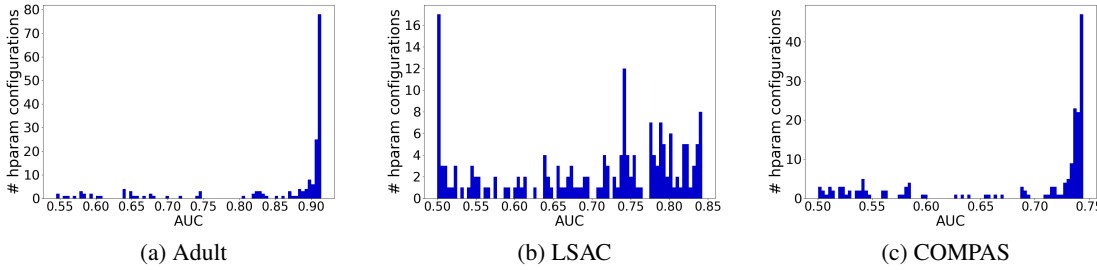

(a) Adult         (b) LSAC         (c) COMPAS

Figure 3: Performance distribution of the grid search. The x-axis shows the AUC score used in the grid search to determine the optimal hyperparameter combination. The y-axis is the count of how many hyperparameter combinations yielded this AUC score.

LSAC, the distribution is wider but there are still several hyperparameter combinations with very good results. So given the deviations that exist between the authors' results and ours, it is not surprising that we also happen to get different hyperparameters. This doesn't necessarily have any deep reason, but might just be due to random differences, as Fig. 3 suggests.

# E   AUC learning curves

Figure 4 shows the AUC learning curves of ARL, IPW(S) and IPW(S+Y) on the validation set of LSAC. As can be seen from the learning curves, IPW(S+Y) consistently outperforms ARL on the LSAC dataset.

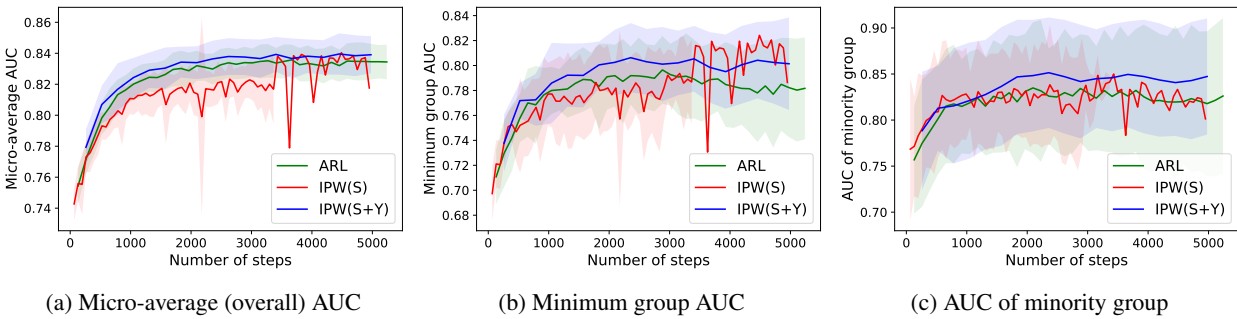

(a) Micro-average (overall) AUC      (b) Minimum group AUC      (c) AUC of minority group

Figure 4: Different AUC values during training of ARL, IPW(S) and IPW(S+Y) on the validation set of LSAC. Average and standard deviation over 10 random seeds.

# F   Results for 100k maximum training steps

As can be seen from Tables 11 and 12, changing the maximum number of training steps from 5k to 100k did not change performance significantly. We did not have enough computational resources to train the DRO model on the Adult dataset, but given the rest of the results it seems unlikely it would have yielded different results.

| Dataset | Method | AUC avg | AUC macro-avg | AUC min | AUC minority | Accuracy |
|---|---|---|---|---|---|---|
| Adult | baseline | **0.9106 ± 0.0009** | **0.9197 ± 0.0015** | **0.8854 ± 0.0010** | **0.9430 ± 0.0042** | 0.8523 ± 0.0049 |
| Adult | DRO | - | - | - | - | - |
| Adult | ARL | 0.9098 ± 0.0007 | 0.9182 ± 0.0006 | 0.8852 ± 0.0009 | 0.9400 ± 0.0025 | **0.8540 ± 0.0020** |
| LSAC | baseline | **0.8370 ± 0.0037** | **0.8295 ± 0.0044** | **0.8151 ± 0.0045** | **0.8347 ± 0.0086** | **0.8655 ± 0.0036** |
| LSAC | DRO | 0.8291 ± 0.0043 | 0.8219 ± 0.0048 | 0.8084 ± 0.0063 | 0.8284 ± 0.0148 | 0.8587 ± 0.0041 |
| LSAC | ARL | 0.8208 ± 0.0052 | 0.8130 ± 0.0058 | 0.7983 ± 0.0067 | 0.8227 ± 0.0087 | 0.8025 ± 0.0147 |
| COMPAS | baseline | 0.7356 ± 0.0024 | 0.7340 ± 0.0030 | 0.6998 ± 0.0027 | **0.7552 ± 0.0071** | 0.6731 ± 0.0054 |
| COMPAS | DRO | 0.7361 ± 0.0031 | **0.7348 ± 0.0032** | 0.7004 ± 0.0028 | 0.7509 ± 0.0101 | 0.6697 ± 0.0075 |
| COMPAS | ARL | **0.7364 ± 0.0026** | 0.7347 ± 0.0032 | **0.7008 ± 0.0023** | 0.7481 ± 0.0080 | **0.6756 ± 0.0091** |

Table 11: Main results for 100k max steps: Baseline vs ARL vs DRO, best results for each dataset are marked bold

| Dataset | Method | AUC avg | AUC macro-avg | AUC min | AUC minority | Accuracy |
|---|---|---|---|---|---|---|
| Adult | ARL | 0.9098 ± 0.0007 | 0.9182 ± 0.0006 | 0.8852 ± 0.0009 | 0.9400 ± 0.0025 | 0.8540 ± 0.0020 |
| Adult | IPW(S) | 0.9087 ± 0.0005 | 0.9181 ± 0.0007 | 0.8826 ± 0.0008 | 0.9437 ± 0.0012 | **0.8552 ± 0.0010** |
| Adult | IPW(S+Y) | **0.9108 ± 0.0007** | **0.9209 ± 0.0015** | **0.8854 ± 0.0008** | **0.9471 ± 0.0045** | 0.7499 ± 0.0155 |
| LSAC | ARL | 0.8208 ± 0.0052 | 0.8130 ± 0.0058 | 0.7983 ± 0.0067 | 0.8227 ± 0.0087 | 0.8025 ± 0.0147 |
| LSAC | IPW(S) | 0.8202 ± 0.0041 | 0.8158 ± 0.0046 | 0.7982 ± 0.0049 | 0.8222 ± 0.0090 | **0.8543 ± 0.0029** |
| LSAC | IPW(S+Y) | **0.8369 ± 0.0037** | **0.8288 ± 0.0046** | **0.8139 ± 0.0046** | **0.8336 ± 0.0074** | 0.8367 ± 0.0088 |
| COMPAS | ARL | **0.7364 ± 0.0026** | **0.7347 ± 0.0032** | **0.7008 ± 0.0023** | **0.7481 ± 0.0080** | **0.6756 ± 0.0091** |
| COMPAS | IPW(S) | 0.7312 ± 0.0020 | 0.7303 ± 0.0027 | 0.6918 ± 0.0029 | 0.7423 ± 0.0086 | 0.6670 ± 0.0086 |
| COMPAS | IPW(S+Y) | 0.7348 ± 0.0034 | 0.7327 ± 0.0039 | 0.6979 ± 0.0033 | 0.7458 ± 0.0079 | 0.6701 ± 0.0062 |

Table 12: Comparison ARL vs IPW for 100k maximum training steps.

# G    Extended Experiments on Image Classification

In order to investigate the performance of ARL on our custom image dataset, we ran a series of different experiments. However, drawing definite conclusions from the obtained results and proposing alterations is beyond the scope of this paper and is thus left as future work.

## G.1    Dataset Details

For keeping the classification task binary, all original class labels larger than 23 were given the new label 1 and all other original class labels were given the new label 0. For each new label, half of the original classes were assigned as belonging to a protected group. To introduce noise, this re-assignment happens with a probability of 0.9. Samples with original labels that are not part of this re-assignment have a probability of 0.1 of becoming a member of a protected group. To eschew the ratio of protected/non-protected samples, samples that become a member of a protected group are discarded from the dataset with a certain probability. In the dataset EMNIST_35, this probability is 0.5 and in the dataset EMNIST_10, this probability is 0.9. This results in about 10% of the samples belonging to the protected group in EMNIST_10 and about 35% of the samples belonging to the protected group in EMNIST_35. The new binary labels are balanced with about 48% of the samples belonging to the new class 1 in both datasets. The dataset reflects that non-noise members of protected groups can be structurally different from non-members with regards to their features. For EMNIST_35, the training set contains approximately 84000 samples and the test set contains approximately 14000 samples while for EMNIST_10, the training set contains approximately 61000 samples and the test set contains approximately 10000 samples.

## G.2    Different Adversary Architectures

We tested a total of three different architectures for the ARL adversary. All consist of a single convolutional layer with $3 \times 3$ kernels and stride 1, a max-pooling layer with a $2 \times 2$ kernel and stride 1 and a subsequent single linear layer that directly maps the flattened output of the previous layer with appended target label to the scalar output. This setup was chosen in order to stay close to the original architecture for tabular data, as proposed by Lahoti et al. The model *ARL_strong* uses 64 filters, the model *ARL* uses 32 filters and the model *ARL_weak* uses 2 filters in the convolutional layer. The learner and the baseline are kept as described in Section 4.2.

| Dataset | Method | AUC avg | AUC macro-avg | AUC min | AUC minority | Accuracy |
|---|---|---|---|---|---|---|
| EMNIST_35 | baseline | **0.9507 ± 0.0037** | **0.9436 ± 0.0041** | **0.9256 ± 0.0055** | **0.9256 ± 0.0055** | **0.8778 ± 0.0059** |
| EMNIST_35 | ARL | 0.9370 ± 0.0044 | 0.9281 ± 0.0052 | 0.9051 ± 0.0072 | 0.9051 ± 0.0072 | 0.8616 ± 0.0057 |
| EMNIST_35 | ARL_weak | 0.9354 ± 0.0072 | 0.9260 ± 0.0082 | 0.9017 ± 0.0112 | 0.9017 ± 0.0112 | 0.8587 ± 0.0095 |
| EMNIST_10 | baseline | **0.9624 ± 0.0013** | **0.9170 ± 0.0022** | **0.8645 ± 0.0040** | **0.8645 ± 0.0040** | **0.8983 ± 0.0025** |
| EMNIST_10 | ARL | 0.8709 ± 0.0696 | 0.7873 ± 0.0875 | 0.6881 ± 0.1117 | 0.6881 ± 0.1117 | 0.7924 ± 0.0731 |
| EMNIST_10 | ARL_weak | 0.9524 ± 0.0032 | 0.8983 ± 0.0077 | 0.8356 ± 0.0132 | 0.8356 ± 0.0132 | 0.8836 ± 0.0055 |

Table 13: Extended experiments: Baseline vs ARL on image data, best results for each dataset are marked bold.

| Dataset | Method | batch size | learning rate |
|---|---|---|---|
| EMNIST_10 | baseline | 512 | 0.01 |
| EMNIST_10 | ARL_weak | 256 | 0.01 |
| EMNIST_10 | ARL | 256 | 0.1 |
| EMNIST_10 | ARL_strong | 512 | 0.01 |
| EMNIST_35 | baseline | 512 | 0.01 |
| EMNIST_35 | ARL_weak | 512 | 0.01 |
| EMNIST_35 | ARL | 512 | 0.01 |
| EMNIST_35 | ARL_strong | 512 | 0.1 |

Table 14: Hyperparameters with the best overall AUC for each dataset and training method. Models were trained for 5k steps or until the early stopping criterion was reached.

## G.3 Training Procedure and Optimal Hyperparameters

The learning rate and batch size of all image models were determined by performing a grid search over the same hyperparameter space as defined for the other datasets in Appendix C. The grid search procedure is the same as in Section 2.3 but with reserving 10% of the training set as validation set instead of performing cross validation. For all image models, the hyperparameters are listed in Table 14. All image models were trained using the AdaGrad optimizer and all reported results are obtained from models that were trained for 5k steps or until the early stopping criterion, that is the overall AUC on the validation set had not improved for 10 epochs, is reached.

## G.4 The Effects of Different Datasets

Table 13 shows the extended results for ARL on the different datasets. In our experiments on image data, the claim of Lahoti et al. that ARL improves the performance on the worst-off group could not be confirmed to hold. Rather, ARL leads to an even lower performance on the minority sub-group which stands in contrast to the original motivation of ARL, that is to increase the performance on the worst-off group. This is independent from the ratio of protected to non-protected samples in the dataset as it happens in both EMNIST_35 and EMNIST_10. It is notable that ARL sees a large performance drop over ARL_weak on EMNIST_10 but not on EMNIST_35.

## G.5 Computational Identifiability

Similar to Section 3.1, we tested whether the adversary is in principle capable of predicting the protected group in a supervised setting. Table 15 shows the accuracies that the different settings for ARL achieve on this task. Due to computational constraints, the reported numbers are results from a single training run of 5000 steps using the hyperparameters that were obtained in the grid search of the complete learner/adversary setting. ARL has major difficulties identifying the minority sub-group in the strongly imbalanced dataset EMNIST_10. The performance on the minority sub-group increases with increasing representation in the dataset as the results on EMNIST_35 show. This is in line with our expectations of a classifier whose optimization target is to improve the overall performance. Further, increasing the complexity of the adversary heavily increases the performance on the minority sub-group while the overall performance and performance on the majority sub-group is only slightly reduced.

|          | EMNIST_10 | | | EMNIST_35 | | |
|----------|-----------|------|----------|------------|------|----------|
|          | ARL_strong | ARL | ARL_weak | ARL_strong | ARL | ARL_weak |
| Overall  | 0.91 | 0.91 | 0.91 | 0.73 | 0.73 | 0.71 |
| Majority | 1.00 | 1.00 | 1.00 | 0.87 | 0.89 | 0.90 |
| Minority | 0.00 | 0.01 | 0.00 | 0.47 | 0.43 | 0.34 |

Table 15: Accuracies of predicting the protected sub-group in a supervised setting.

