# OpenReview forum: "Replication: Fairness without demographics through Adversarially Reweighted Learning"
_ML_Reproducibility_Challenge/2020 — Reject_

### Official Review · AnonReviewer1 · 2021-02-28
**Please check the original paper's optimal hyperparameters.**

**Rating:** 5
**Confidence:** 3

**Review:**

The author reimplements the paper "Fairness without demographics through Adversarially Reweighted Learning".

On one hand, the good news is the author reproduces ARL's performance in the original paper. The frustrating news is that the author finds the baseline DRO's performance is much higher compared to that originally reported.

The author says the original paper does not provide the grid search results. Fortunately, I find that the authors of the original paper publish their code and the optimal hyperparameters in their [GitHub repository](https://github.com/google-research/google-research/tree/master/group_agnostic_fairness).

The optimal hyper-parameters from this paper's Appendix B are very different from the original paper's grid search result (see GitHub).

So I think the author should revisit the experiment and try to give some insight on what are the optimal hyperparameters.

**Familiar With The Original Paper:**

I have not read the original paper

**Reproducibility Summary:**

Report has summary

---

> ### Author Response · Authors · 2021-03-17
> **Hyperparameters released after deadline, we will include new experiments and discussion in final report**
>
> Thank you for pointing out the hyperparameters in the paper's Github repository. These were only released after the MLRC deadline (see the commit history on Github), which is why we didn't address them in our report. We are now re-running the experiments with those hyperparameters and will shortly include the results in our report, including a discussion of the difference compared to the optimal parameters we found. We hope this fully addresses the concerns you have.

---

> > ### Author Response · Authors · 2021-03-21
> > **Many different hyperparameter combinations perform close to optimally**
> >
> > As a follow-up to our previous response: we have done two experiments using the now published hyperparameters, first running the authors' code with their hyperparameters (Appendix A) and secondly comparing their optimal hyperparameters to ours (Appendix D).
> > Interestingly, we find that there is a wide range of hyperparameters with close to optimal performance, and using the authors' hyperparameters works almost as well as using our own. This means that the difference in hyperparameters is probably a simple consequence of other slight implementation differences (such as our use of early stopping) or random fluctuations. We have also improved the Discussion, which gives more details on these differences.
> >
> > Finally, we noticed through the newly released list of hyperparameters that the authors tune the learning rates for adversary and learner independently, this was not clear to us from the paper itself. We have rerun our own grid search using this methodology and updated all results accordingly, but this resulted in no qualitative changes.

---

### Official Review · AnonReviewer2 · 2021-03-08
**A good reproducibility paper with additional applications, but not completely consistent with original paper**

**Rating:** 6
**Confidence:** 4

**Review:**

This manuscript provides a pytorch implementation about the paper "Fairness without demographics through adversarially reweighted learning". The authors do not perfectly reproduce the results, but the trend seems to be consistent with the original paper. Hyperparameter tuning is sufficient. Additional contribution of this paper is to apply this method to image data.

**Familiar With The Original Paper:**

I have read the original paper

**Reproducibility Summary:**

Report has summary

---

> ### Author Response · Authors · 2021-03-17
> **Discrepancy is mainly due to better baseline performance in our results, discussion of potential causes**
>
> Thank you for taking the time to review our report. Regarding the inconsistencies with the original paper: as we discuss, the main difference is that our baselines perform slightly better than the original authors', which negates the advantage of ARL. We discuss some potential reasons for this in the report, such as the fact that the fixed number of training steps used by the authors may be suboptimal for some methods, whereas we use early stopping. We also show (see Table 6) that our PyTorch implementation tends to perform slightly better than the original Tensorflow implementation (with identical hyperparameters), which in our view makes implementational errors unlikely.
>
> Finally, the original authors released their optimal hyperparameters soon after the MLRC deadline; we are now re-running experiments using those, which will allow us to narrow down potential sources of the discrepancy even more, and we will update our report shortly.
>
> Hopefully, this addresses any concerns you might have about the discrepancy between our results and those from the paper. If you have additional suggestions on what we should do differently or potential causes we should investigate, please let us know.

---

> > ### Author Response · Authors · 2021-03-21
> > **Included better discussion of discrepancy, ran experiments with newly released hyperparameters**
> >
> > As a follow-up to our previous response: we have improved the Discussion to better highlight our hypotheses for why ARL does not show any advantage in our results and why our baselines are stronger than those in the original paper. We have also included experiments with the newly released hyperparameters (see Appendices A and D), which however don't change our conclusion.
> >
> > Finally, we noticed through the released list of hyperparameters that the authors tune the learning rates for adversary and learner independently, this was not clear to us from the paper itself. We have rerun our own grid search using this methodology and updated all results accordingly, but this resulted in no qualitative changes.

---

### Decision · Program_Chairs · 2021-03-31

**Decision:**

Reject

**Comment:**

Overall reviews and/or the paper content not good enough for the AC to recommend to the journal.